# Advancing aphasia psychological care in Ireland: A participatory study with people with aphasia and clinicians

Molly X. Manning[1,2]*, Brian McClean[3], Libby Cunningham[4], Jolene Dervin[5], Marcia Ward[6], Aoife Carolan[7]

**1** School of Allied Health, University of Limerick, Co., Limerick, Ireland, **2** Health Research Institute, University of Limerick, Co., Limerick, Ireland, **3** Acquired Brain Injury Ireland, Teach Fáilte, Mountbolus, Tullamore, Co., Offaly, Ireland, **4** Mater Misericordiae University Hospital, Eccles Street, Dublin, Ireland, **5** Health Service Executive, CHO8, Ireland, **6** Cork University Hospital, Wilton, Cork, Ireland, **7** Health Service Executive, ECCN 10 (CHO 8), Ireland

* molly.manning@ul.ie

## Abstract

### Background

Aphasia increases the risk of mental health issues, yet psychological care for aphasia (APC) is not routinely available. Given the significant psychosocial impact, APC requires input from multiple disciplines and settings, and necessarily responds to a person's needs, which shift over time. As a complex intervention, it is important that service providers, users, and policymakers are involved in developing effective and contextual implementation strategies. In this study, the aim was to meaningfully engage people with aphasia and inter-disciplinary aphasia clinicians to create a shared vision of ideal APC and to identify context-specific implementation considerations in Ireland.

### Methods

A series of interdisciplinary clinician research workshops and parallel aphasia Public and Patient Involvement (PPI) meetings were convened. Participatory Learning and Action (PLA) tools and techniques supported inclusive research spaces; and discussions were scaffolded by constructs from the implementation science framework, Normalisation Process Theory (NPT). PPI contributors inputted into planning the clinician workshops and reviewed and commented on the discussion summaries.

### Results

Key principles for APC and Ireland-specific implementation considerations were identified. These included the need for a unified vision, clear clinical roles and referral pathways, improved team structures and resources, clinician training, and active involvement of people with aphasia in APC design.

**Data availability statement:** All relevant data are within the manuscript and its Supporting Information files.

**Funding:** This work was supported by Seed Funding from the Faculty of Education and Health Sciences, University of Limerick, awarded to Dr Molly Manning in 2021. The funders had no role in study design, data collection and analysis, decision to publish, or preparation of the manuscript.

**Competing interests:** The authors have declared that no competing interests exist.

## Conclusions

Through NPT and PLA informed workshops, people living with aphasia and clinicians generated ideas about how an interdisciplinary coordinated model of APC might be developed and implemented. The findings offer early guidance for developing coordinated, interdisciplinary APC in Ireland. The participatory implementation approach may be adapted to other conditions or contexts to support locally relevant intervention planning.

## Introduction

Aphasia is a risk factor for poorer social networks and health-related quality of life and mood disorders like depression and anxiety [1–5]. Compared with others with stroke, people with aphasia are three times more likely to experience agitation and four times more likely to experience irritability [6]; and are at higher risk of depression, anxiety and psychosis 1-month post-stroke [7]. A meta-analysis of 128 studies with over 15,000 participants with stroke reported a 50% higher risk of depression in people with versus without aphasia and a prevalence of any depressive disorder of 52.2% (95% CI = 34.9% to 69.3%) versus 33.5% (95% CI = 30.3% to 36.8%) [8]. Aphasia is also associated with more severe and more persistent depression post-stroke [1]. The incidence of post-stroke anxiety is about 20–25% [9]; however, this increases to 44% in community-dwelling people with aphasia [10].

Given these pervasive and complex psychosocial ramifications, Aphasia Rehabilitation Best Practice statements advocate support with mental health, psychological and social wellbeing and community reintegration, including from Speech and Language Therapists (SLTs), Psychologists (PSYs), and community-based organisations and aphasia groups, as appropriate [11]. A growing body of research is examining the feasibility and effectiveness of a range of aphasia-adapted psychological therapeutic approaches including Cognitive Behavioural Therapy [12], Acceptance and Commitment Therapy [13], Behavioural Activation [14], Motivational Interviewing [15], Solution-Focused Brief Therapy [16], and counselling training and education for SLTs working with people with aphasia [17]. There is attention too to facilitating the coordinated and synergistic working of various professionals that might be involved in providing therapeutic psychological support. A stepped psychological care approach is recommended in stroke clinical guidelines, with input from Occupational Therapists (OTs), PSYs, and other professionals [18]. Stepped care is a well-established framework of coordinating support across multiple service settings and professionals in such a way that patients obtain the type of support that they need at different times in their lives. It is underpinned by the stepped care paradigm in mental health, in which patients are matched with the appropriate amount and intensity of support needed at any point. Despite variation in the way that stepped care is implemented, its effectiveness in populations with depression, anxiety and post-traumatic stress disorder has been demonstrated through umbrella review [19].

The stepped care approach has recently been specified for post-stroke aphasia by Baker [20]. This framework articulates the interdisciplinary input required and possible intervention approaches across a hierarchy of 4 increasingly specialist and/or intensive levels of support [20]. This is important given the shared remit of SLT, PSY, and related disciplines in supporting psychological wellbeing, leveraging complementary expertise in supported communication and psychological therapies. A range of universal psychosocial support interventions are specified at Level 1, including peer befriending and self-management support, as well as mood screening. More focused support is available across levels 2 (which can be SLT-led with support from other mental health disciplines) to levels 3–4 (mental health and psychology-led, enabled by SLT communication support).

Shortcomings in aphasia care are widely reported by clinicians and people living with aphasia, however [21–24]. There is evidence that healthcare professionals may also experience organisational barriers supporting people with aphasia and in enacting communication strategies. As a result, they may avoid or dread speaking with patients with aphasia, despite wanting to help. Ongoing training and support is required, as well as cultural and organisational change to support patient-provider communication with people with aphasia in the ward [25]. As identified by Baker and colleagues in qualitative focus groups with multidisciplinary stroke clinicians, barriers and enablers to implementing stepped aphasia psychological care include a lack of clinician experience and specialist training, insufficient access to psychologists and interdisciplinary working, and a lack of leadership, financing and resourcing [26]. There is also a need to develop and test the psychometric robustness of non-verbal self-report measures of depression and suicidality in aphasia [27]. As a complex intervention, comprising multiple component interventions, disciplines and care settings, a stepped aphasia psychological care framework will ideally be shaped to local contexts with meaningful engagement of all interest holders throughout its development and implementation [28].

The focus of this study was to explore how such a framework might be adapted for the Irish population and healthcare context. Stepped psychological care is well-established in Ireland and underpins national mental health service vision and strategy [29,30]. More recently, it has been recommended in national clinical guidelines for stroke [18]. Ireland's National Stroke Strategy 2022−27 champions increasing the availability of psychologists and ensuring that access to stroke care and psychological support is equitable for people with aphasia [31]. Shortcomings in access to psychologists and longer-term community support for people with stroke are well-documented, however [32,33], and rates of mood screening in stroke are low [34]. Mirroring Baker and colleagues' findings from Australia, research with interdisciplinary clinicians in Ireland and the United Kingdom also highlights a perceived lack of synergistic working between SLTs, PSYs and other aphasia professionals in supporting emotional wellbeing with aphasia [35–38]. SLTs specifically report a lack of training and confidence in delivering psychosocial support as well as a lack of a coordinated pathway of care [35,38,39]. PSYs and OTs report using varied communication techniques with people with aphasia, but are less confident supporting more complex conversations in the context of delivering psychological care [37].

In this participatory implementation study, aphasia clinicians and people with aphasia were engaged in discussions on the potential for developing and implementing an improved framework of aphasia psychological care in Ireland. The methodological approach sought to synergistically combine the participatory health research methodology, Participatory Learning and Action (PLA) [40], with the implementation science framework, Normalisation Process Theory (NPT) [41]. PLA is underpinned by an axiology of meaningful community involvement and ownership in the research process [42]. A range of methods, tools and techniques may be applied to support co-researchers to share, reflect, investigate and apply their lived knowledge and experiences to action change. PLA flexibly incorporates visual and creative modalities and group-based methods and has been applied to engage diverse communities, including those from diverse language and culture backgrounds in research about migrant health, as well as in research with aphasia co-researchers [43–45].

NPT draws on sociological theory and empirical evidence to articulate the ways in which complex interventions become adopted and embedded into routine clinical practice [46,47]. The most recent NPT coding manual specifies 3 key domains, Context, Mechanisms, and Outcomes, each of which containing constructs relating to the implementation

environment; the 'work' that people do, individually and collectively, to adopt, enact and sustain complex interventions; and the practical, relational and structural outcomes of implementation [48].

This combined PLA / NPT methodology has been described previously, including in the context of research examining culturally responsive practice, refugee and migrant healthcare [49]. It harnesses the rigour of PLA in equalising research involvement and knowledge generation across diverse non-academic and community partners, and which has been used successfully in the context of aphasia [45,50]; whilst leveraging tested ideas from implementation science to enhance the applicability and usefulness of the outputs generated [48].

The purpose of this study was to convene a participatory research space with interdisciplinary *aphasia clinicians*, guided by discussions with *aphasia Public and Patient Involvement (PPI) contributors*, to consider: [1] the need for improved aphasia psychological care in Ireland; [2] the scope of aphasia psychological care for the purpose of this study; [3] characteristics of ideal aphasia psychological care; [4] the utility of a stepped care approach; [5] training and support needs for clinicians; and [6] implementation factors specific to the Irish context.

## Materials and methods

The reporting of this study follows Standards for Reporting Qualitative Research (SRQR) guidance [51] (S1). Ethical approval was obtained from the Faculty of Education and Health Sciences Research Ethics Committee at the University of Limerick (2021_12_19). A series of research meetings were convened separately with aphasia clinician co-researchers and aphasia PPI contributors. These were facilitated by the first author, an experienced PPI and participatory research facilitator and academic SLT.

### Clinician co-researchers

The first and final authors established and co-led SEA (Supporting Emotions with Aphasia), an Irish interdisciplinary community of practice comprising mostly SLTs, OTs and PSYs. They identified potential clinical co-researchers who were members of this network and/or had presented to the network previously, and who would bring interdisciplinary perspectives across the continuum of aphasia care (i.e., acute, early supported discharge, primary care, charitable sector). The first author subsequently contacted each person, circulated study information and obtained written informed consent between 10–25 March 2022. The clinician co-researchers each brought over 10 years' experience of working with people with aphasia. They included 2 Senior SLTs working in community and inpatient rehabilitation settings for the Irish health service, Health Service Executive (HSE) (fourth and final authors); a Clinical Psychologist working in a third sector support organisation (second author); a Clinical Specialist Occupational Therapist in Stroke Care (third author) working in Early Supported Discharge (ESD); and a Clinical Neuropsychologist (fifth author), working in stroke care in an acute teaching hospital.

Four online research workshops, each 1.25 hours, were conducted with the clinical co-researchers. Their roles were to agree the scope of aphasia psychological care (APC) for the purpose of the study, reflect upon current service provision, generate principles of ideal APC, consider the utility of a stepped care approach as well as implementation considerations such as stakeholder identification and information needs. The clinician co-researchers also agreed to be co-authors on the present manuscript.

### Aphasia PPI contributors

The aphasia PPI contributors comprised 3 working-aged adults (2 women and 1 man) living with post-stroke aphasia for approximately 10–15 years. They were experienced PPI contributors, having worked together with the first author on various research projects relating to optimising support for living well with aphasia in the previous 4–5 years. Indeed, much of this prior research underpinned the rationale for the present study [21,39,50,52–54]. The first author circulated the study information and subsequently met each person to obtain written informed consent between 10–31 March 2022.

Subsequently, 4 PPI meetings were convened, each facilitated by the first author. The first 2 were online due to Covid-19 restrictions and lasted 1.5 and 1 hours respectively. This resulted in 1 previous PPI member declining to be involved. The rest were in-person and lasted 3 hours each, including a lunchbreak. Lunch and refreshments were provided, and remuneration was in the form of vouchers. The PPI contributors' roles were to support the planning of clinician workshops, to discuss, comment and extend the knowledge generated by the clinicians, and to validate the final report and discuss dissemination plan. The contributors were offered but declined co-authorship on this manuscript as they wanted to maintain their anonymity.

### Operationalisation of NPT constructs

The focus of the clinician meetings and discussion prompts were underpinned by key domains and constructs from NPT [48]. These were operationalised for the purpose of the study by the first author and were used flexibly to guide knowledge sharing and generation relating to APC in Ireland. An indicative outline of how the NPT constructs guided discussions is in Table 1.

### Sequencing of meetings

The sequencing of meetings is shown in Table 2. At an initial meeting with aphasia PPI contributors, the plan for the clinician meetings was discussed. Two meetings with clinician co-researchers followed, focusing on current service provision, the scope of APC in the project, and principles of ideal APC. Following these, 2 meetings with aphasia PPI contributors provided an update on clinician meetings and opportunities for discussion and comment. The final 2 meetings with clinician co-researchers included a presentation of PPI contributor comments and discussions about stepped care and implementation considerations in Ireland. In a final PPI meeting, discussions spanned a summary of clinician meetings, dissemination planning, and future research. There was full attendance at all meetings.

### Knowledge generation and synthesis

Two 'living' slide decks were created and maintained by the facilitator (first author), 1 per group, including an aphasia-friendly slide deck for the PPI meetings [55]. These outlined the meeting agenda and helped to provide a structure and focus to shape the discussions, to summarise and record all knowledge generated, for example, by incorporating

---

**Table 1. Operationalisation of NPT constructs.**

**Mechanisms**

**Coherence**
• **Differentiation:** Clearly identifying problems within current services that APC aims to address.
• **Individual Specification:** Drawing on personal experiences and knowledge relevant to APC.
• **Communal Specification:** Developing a shared understanding of APC's purpose and core components.
• **Appraisal:** Recognising the value and necessity of APC and a stepped-model approach.

**Cognitive Participation**
• **Initiation:** Defining who will lead APC implementation and foster engagement. Considering stakeholder interests and communication strategies to maximise buy-in and support.

**Collective Action**
• **Interactional Workability:** Assessing clinicians' confidence in their ability to deliver APC in Ireland.
• **Skillset Workability:** Identifying training and knowledge requirements for aphasia clinicians to implement APC effectively.
• **Contextual Integration:** Evaluating organisational and employer support needed for APC implementation.

**Context**
Reflecting on current service provision and resource constraints in Ireland, alongside population health factors such as stroke and aphasia incidence.

**Table 2. Sequence and focus of meetings.**

| Meeting | Date | Input Materials and Discussion Prompts |
|---|---|---|
| PPI meeting 1 | March 2022 | Accessible presentation with project overview, recap on relevant aphasia literature and studies the group had already been involved in, and plan for clinician meetings. |
| Clinicians meeting 1 | April 2022 | Presentation with information about (1) the study; (2) relevant literature on living well with aphasia, stepped psychological care, and current services.<br>Sample prompts: Does this information match with your experiences working in aphasia psychological support? What, if any, issues need to be addressed? |
| Clinicians meeting 2 | May 2022 | Updated slide deck including all miro.com contributions and a summary prepared by the facilitator for discussion and approval.<br>Sample prompts:<br>(1) What is in and out of scope across patient focus; pathway start/finish; emotional support; care settings and services?<br>(2) What would an ideal model of emotional/psychological aphasia support look like? For example: key principles, team structures, mentorship and support, training, resourcing and financing, management and policy support. |
| PPI meetings 2–3 | May 2022 | Updated accessible slide deck with information about the clinicians and knowledge generated, including four themes of clinician comments: (a) a better pathway of care; (b) people who can help; (c) training and support for clinicians; (d) the focus of emotional care. |
| Clinicians meeting 3 | June 2022 | Updated slide deck including PPI contributor comments.<br>Sample prompts (re stepped care): Is it useful? Does it cover our scope? How might it work in our context? Is it feasible? What is already being done? How could it interface with other priorities? |
| Clinicians meeting 4 | June 2022 | Updated slide deck and a summary of knowledge generated in earlier sessions.<br>Sample prompts: What are priorities? Who are key interest holders? What information do they need and in what format? |
| PPI meeting 4 | May 2023 | Updated accessible slide deck summarising key points of clinician meetings. |

screenshots of miro.com comments to serve as a practical visual representation, and to facilitate inter-group sharing. Specifically, these provided [1] background information on the project and on relevant aphasia literature; [2] discussion prompts relevant to the specific meeting; [3] visual recording of ideas and comments generated in previous meetings; [4] iterative synthesis and summary of knowledge generated in previous meetings; [5] inter-group comments and contributions (e.g., the clinicians were presented with PPI contributions at their third meeting; PPI contributors were presented with an accessible overview of clinicians' discussions in PPI meetings 2–4). An overview of materials and sample discussion points is in Table 2.

The approach to data generation and synthesis was informed by PLA techniques including *listing* (i.e., capturing discrete comments and ideas visually on stickies in response to a discussion prompt), and *card-sorting* (i.e., visually re-organising stickies, for example by *combining* and/or *comparing* content) [56]. Adapted PLA techniques have been applied previously including with people with aphasia and clinicians [49,57]. In response to prompt questions (see Table 2 for examples), discussion points were recorded in real-time on stickies (virtually on miro.com for online meetings). The clinicians opted to create their own stickies within a miro.com whiteboard, which was then left 'open' for comment for several days, which enabled further reflection and contribution. The aphasia PPI contributors agreed that the facilitator would record their comments on stickies – whether virtually or physically – and then read them back to check accuracy and

completeness. At each meeting, the facilitator led the process of grouping stickies with similar or related content, and then created a summary of the content, which would be presented and discussed at the subsequent meeting. The final version of each slide deck was circulated to members of both groups by email.

Although NPT constructs guided the discussion prompts, the knowledge generated through PLA techniques was analysed using an iterative thematic analysis process [58]. The first author led this process, coding all material, with continuous scrutiny and collaborative input from co-authors during meetings and through co-creation of the final report. The analysis was inductive and highly iterative, with themes refined through ongoing discussion.

Initially, NPT informed the development of questions related to clinicians' sense-making, their perceptions and experiences on current care (Differentiation, Individual Specification). After each clinician meeting, all contributions and virtual 'stickies' were systematically reviewed to identify similarities, differences, and emerging insights. These informed the development of preliminary themes and descriptors capturing patterns of meaning relevant to the research questions.

A summary of these evolving themes, supported by all contributions, was presented at the start of each subsequent meeting, functioning as a de facto data analysis clinic This approach enabled member checking and strengthened trustworthiness [59]. The emerging themes guided further discussion about principles of good care should entail, and the perceived value and differences compared with current practice (Communal Specification and Appraisal).

The knowledge generated through these discussions was again synthesised and thematically organised. Refined themes were subsequently presented to guide dialogue on stepped-care design and priorities for implementation (Cognitive Participation and Collective Action constructs). Following the second clinician meeting, these evolving themes were also presented to PPI contributors, whose input was incorporated alongside clinician comments to validate findings and inform subsequent meetings.

To enhance reflexivity, the first author maintained detailed notes, documenting emerging interpretations, positionality, and potential biases [60]. A de-identified version of the final clinician slide deck—incorporating all contributions and the outputs of the iterative thematic analysis—is provided in Supplementary S2 File. Feedback from this final stage was thematically summarised, forming the basis of the final reported themes.

## Results

The knowledge generated is summarised below including a perceived need to improve APC in Ireland, the scope of APC for this study, a stepped care approach to delivering APC, clinician training and support needs, and implementation factors specific to the Irish context. Illustrative content from both groups is provided.

### The need to improve APC in Ireland

The clinicians identified and reflected on shortcomings with current service provision, including a lack of availability and access to, psychological support; no specialist pathway of aphasia care; a lack of comprehensive data tracking to monitor access to care; and geographical variation in terms of community support teams and ESD. Inconsistency in access to support was also emphasised by PPI contributors. The clinicians perceived a lack of onward referral pathways when discharging patients from hospital, long inpatient rehabilitation waiting lists, and an overly medicalised approach to stroke and brain injury. Finally, they noted a lack of shared conceptual understanding of the purpose and distinct nature of APC among health professionals. They agreed on the need for aphasia health professionals to have appropriate training and opportunities for interdisciplinary working and support.

### APC scope in this study

The clinicians agreed APC would span emotional, social and cognitive support for people with acquired, non-progressive aphasia from brain injury onset across the continuum of care. A summary is in Table 3.

**Table 3. Agreed scope of APC in this study.**

| Focus | Description |
|---|---|
| Patient Focus | All individuals with aphasia secondary to acquired brain injury including people with pre-existing emotional, mood and/or psychiatric conditions. Excludes individuals with Progressive aetiologies including Primary Progressive Aphasia; acquired communication impairment with no aphasia; and family members as direct recipients of support. |
| Therapeutic focus | Therapeutic input (direct / indirect) targeting improved psychological wellbeing, mood, cognition and /or social participation. |
| Pathway | From brain injury onset, following patient across continuum of care, as needed. Excludes support prior to hospital admission and/or brain injury diagnosis. |
| Care setting and service | No limiters (e.g., acute, early supported discharge, inpatient and outpatient rehabilitation, third sector support organisations). |

## APC values and ethos

The clinicians agreed on a need for a shared conceptual understanding of the overall purpose of APC that would underpin policy and practice across all care settings. Equity and consistency of access were emphasised, as well as a need for responsive, flexible timing and self-referral options *("no matter where you live…you are entitled to equality of access to effective therapy"; "service needs to be structured to [sic] flexible enough for service users to use service across stroke care continuum").* These points were echoed in the PPI meetings *("long-term and self-referral good ideas").* One PPI contributor highlighted the non-linear nature of adjustment and how people with aphasia might want professional support in the longer-term (*"of all the things that's really great. You can do courses etc., but if you get stuck and want to 'back off' a little, it's a great thing to be able to talk to them about anything – e.g., your speech, your head").*

The clinicians agreed on a need to re-frame the values and focus of care, shifting away from a medical model emphasis on professional input and full clinical recovery, to facilitating wellbeing and health in the context of disability, and normalising and validating distress (*"we need to become more comfortable letting people "be" and not always focusing on the "fix'"; "need to begin with the end in mind, living well with aphasia is not a post therapy idea").*

Synergies with the Personal Recovery [61] paradigm were noted. These included conceptualising 'personal' recovery as distinct from 'clinical' or 'medical' recovery; personally defined; and non-linear, in that people may have periods of wellbeing and deterioration across time, with shifting support requirements. Personal recovery may take place within or outside of the context of formal support services; and thus, recognises individuality, autonomy and personhood. The clinicians agreed on a need to reframe an over-emphasis on the importance of 'services' and professionals in supporting living well (*"more profs not the answer… we are sign-posters not the destination").* This was echoed in PPI meetings (*"have to live life as well").* The clinicians noted a more fundamental question about how the conceptualisation of disability underpins service design. Focusing on medical recovery may in fact impede personal recovery and re-engaging with life. The PPI contributors welcomed the need for a conceptual shift towards responsive and individualised support (*"they need to learn to sit down and listen to the person…what can I do to help you?"),* one which recognises and bolsters the agency and work that people with aphasia do in managing their self-care and adjustment (*"lots of things people can be doing to help themselves").*

Finally, there was agreement among both groups that support for families should be embedded into a model of APC, for example, family peer support groups and tailored support for children of people with aphasia. There was acknowledgement that that family support needs may increase over time. Supporting psychological wellbeing in the context of a family unit was perceived as potentially more effective than working with individuals.

## A stepped care approach

The clinicians broadly agreed that a stepped care approach to APC had potential to provide timely, responsive care over time; but that further specification on care coordination, referral pathways, and roles would be required to make it work in practice. Importantly, they believed that the 'shape' of stepped care should be further examined and conceptually refined. For example, they highlighted a need for a 'level 0' to acknowledge the 'work' that people do without formal support. Again, this resonated with a Personal Recovery conceptualisation of wellbeing, recognising the inherent heterogeneity and personhood of people both pre- and post- aphasia onset. Further clarity on the structure of stepped care would be required in future implementation initiatives, for example later levels (3 and 4) were perceived as not clearly delineated, with a particular lack of evidence for level 4 interventions. The utility of differentiating levels 2 and 3 was also queried ("*I am not sure about the difference …and whether it is helpful to distinguish between them*"). The clinicians also noted that some interventions might transcend levels, for example peer-support groups ("*Are levels too 'fixed'? "Groups often work really well when there are guides who have already travelled the path."*").

The clinicians agreed that Level 1 screening should cover cognition, language and mood as an entry point to more detailed assessment and access to later stepped care levels. There was a perceived need for more awareness of the role of SLT in mood screening, assessment and management across multidisciplinary teams*.*

A range of specific intervention approaches were discussed including 'first principle' universal interventions post-brain injury ("*fatigue management, activity scheduling, physical exercise, mindfulness, and participation in everyday living*") and standalone approaches for depression including aphasia-accessible Behavioural Activation and Cognitive Behavioural Therapy. At level 1, the clinicians highlighted the importance of initiatives *"where people can feel joy and purpose"* and to have a choice of low-level interventions. An example given was having access to a schedule of classes and activities, which could be booked on an ad hoc basis. They agreed on the importance of access to aphasia community interventions such as aphasia cafes, as well as self-management and peer support. The PPI contributors also emphasised the importance of meeting others with aphasia for wellbeing and adjustment (*"want people with similar experience to help"; "meeting others living with aphasia very important"; "want someone that has same problems at home"*). The clinicians recommended that 'alternative modality' interventions, such as yoga and creative therapies, should be explored and available at all levels of APC ("*creative therapies are often seen as a luxury … rather than core services. Aphasia should turn this perception inside out... if I cannot talk, access to other forms of expression is essential, and talk therapy is peripheral*"). The restorative potential of creative therapies resonated strongly with the aphasia group (*"creative therapy – great one"; "music always helps with relaxing"; "mindfulness with art…fantastic…arty and quiet"; "can always hear what my mood is through my choice of music"*).

The clinicians identified universal design and adaptation of health communication environments as priority areas, noting possible opportunities to embed actions into ongoing health literacy and access initiatives. They also emphasised the importance of care coordination and continuity ("*continuity of care is crucial – jumping from one team to another not good"; "there needs to be a case coordinator overseeing the entire model of care"*); these were also emphasised by the PPI contributors (*great idea to see same person each time – just like doctor knows you"*).

## Training and support for clinicians

The clinicians agreed that aphasia clinicians should have the opportunity to complete relevant training and to avail of professional supervision and debriefing. Supervision and emotional support for SLTs was felt to be crucial elements of professional self-care, problem-solving, and debriefing about emotionally charged professional encounters. Such support was identified as relatively unavailable, however. Training needs identified included training in psychological strategies for Level 2 interventions such as Cognitive Behavioural Therapy, Behavioural Activation and Motivational Interviewing for non-psychology staff like SLTs; and supported communication education for non-SLTs.

The discussions highlighted the importance of increasing awareness within the multidisciplinary team of the need for SLTs to be involved in mood screening and assessment, and to engage in joint-working and discussions relating to mood with Psychology and OT colleagues. It was also emphasised that there must be recognition of the positive work already being enacted, including by SLTs, who may in fact need reassurance and validation to build confidence in their existing skills and practices (*"Need for recognition that SLTS provide amazing emotional, compassionate care … sometimes it's more reassurance that we are not missing something and confidence in existing skills and practice"*).

To enhance referral accuracy and increase timely access to APC and SLT, the clinicians agreed on a need to increase aphasia awareness across the continuum of care, including nursing homes, Psychiatry of Later Life units, General Practice, acute, rehabilitation and community settings. The importance of interdisciplinary training aligned with universal interventions (e.g., in empathic and active listening, supporting adjustment, and supported communication) was emphasised. The PPI contributors agreed on the importance of clinicians understanding the experience of adjustment and living with aphasia (*"clinicians have to learn themselves too. They're in a bubble"; "clinicians have to 'click' about aphasia"*). They suggested included education sessions with people with aphasia as expert trainers. Interdisciplinary training and co-learning post-qualification was also recommended

## Contextual implementation factors

Ensuring that APC service design responds to the Irish context was an agreed priority for the clinicians. This included, for example, developing a deeper understanding of current service provision and initiatives across care levels and settings. The clinicians considered how elements of APC might be integrated into existing service structures to maximise efficiency and cost-effectiveness. Examples given were to understand the level 1 supports currently available (e.g., through Irish Heart Foundation), and to explore the feasibility of establishing a pathway of care spanning multiple agencies across third sector support organisations, social care and community services, and the HSE. There was also agreement that the output of the present study should inform stroke and psychological care policy.

The potential resourcing implications of stepped care were noted by clinicians (*"need for a lot more Psychology and Neuropsychology manpower though and I fear therefore is unrealistic maybe"*) and PPI contributors (*"how can they take on more?"*). The clinicians believed that increasing the number of aphasia professionals in isolation would not necessarily provide a solution. Instead, they highlighted a need to review team structures and resourcing to enable better use and coordination of clinicians and health workers involved in APC (*"not to be caught up in the "that doesn't exist … as then nothing will change"*). Suggestions included further involvement of rehabilitation assistants, including in the community, protecting specialist resources (e.g., ESD), clarity on the role of stroke liaison support workers, inclusion of an aphasia 'emotional support' champion on stroke and brain injury teams, and looking at extending the remit of Home Help services beyond personal care.

The potential for transdisciplinary working for unlocking resources was discussed (*"80% of what we offer as professionals could be offered by any other professional on the team"*). Such work would necessitate further participatory sessions to develop the 'shape' of a national APC model, with wider stakeholder consultation and PPI engagement at all stages from design through implementation. The clinicians highlighted how a new HSE primary care network manager structure may offer further potential for inter- and trans-disciplinary working. These opportunities were felt to be important due to a lack of consistent encouragement and support from health service management structures to work with other professional disciplines (*"we need Line Management support to work with other professional disciplines – sometimes not encouraged"*).

They additionally emphasised the need to capture valid, clinical, economic and societal *outcome data* relevant to APC, noting that these data are not routinely collected nor available. This must be underscored by improved political awareness and 'visibility' around the need for APC, since invisible needs and impact are hard to quantify.

The clinicians discussed how to raise awareness of the need for enhanced APC to support interest holder investment and buy-in. One clinician noted a need to *"get the gravity and seriousness of this need front and centre with the medics*

*and heads of service".* Multi-level parties were identified including policymakers and service commissioners (HSE, including National Clinical Programme for Stroke, National Rehabilitation Programme, Communications, and Quality and Patient Safety offices, Department of Health and Children, third sector support organisations, National Office of Clinical Audit, and Professional bodies); local service provider organisations, relevant care pathways (Enhanced Community Care) and service-level managers; among multi-disciplinary clinicians and support workers (e.g., Clinical Nurse Specialists, Medical Consultants, General Practitioners, Health and Social Care Professionals, key workers, Stroke Support Coordinators); and people living with aphasia, including families and caregivers. The clinicians identified different informational needs across these diverse interest parties and a need to tailor dissemination outputs accordingly. These included a range of formats like academic papers, accessible presentations, animations, policy briefs, position papers endorsed by professional bodies, and conference presentations.

## Discussion

A series of participatory research meetings was convened with interdisciplinary aphasia clinicians and aphasia PPI contributors. NPT constructs anchored and inspired discussions about the need for enhanced aphasia psychological care, its scope, ideal principles and contextual implementation considerations. The utility of a stepped care approach was accepted in principle, albeit with a need for further discussion and adaptation. Establishing a values-based APC framework was strongly advocated.

The agreed scope for APC spanned emotional, social and cognitive health for people with aphasia of any non-progressive aetiology. This differs from a predominantly stroke-focused aphasia psychological literature [26,38,54] and will require further consideration and attention to adapting and implementing stroke-focused research evidence. Broadening the remit of APC beyond stroke represents a potentially pragmatic response to current service configurations in Ireland, in which aphasia clinicians commonly cater to a mixed brain injury population, and overall, compared with countries with bigger populations, the prevalence of stroke and aphasia is relatively low [31]. Therefore stepped psychological support in the context of aphasia will necessarily dovetail with a number of synergistic national health policies beyond stroke, including, Sláintecare, which advocates universally accessible, community-centric and integrated health and social care services [62]; and the yet un-implemented Neurorehabilitation Strategy, which aims to maximise health outcomes through integrated, community-centric, person-centred support at the lowest possible level of complexity [63]. Furthermore, the decision to include cognition and psychosocial wellbeing in the scope of the pathway was influenced by the multi-faceted roles of PSYs, OTs, and SLTs working with people with aphasia and brain injury in Ireland. The overlap and synergies with existing neuropsychological rehabilitation pathways should be examined further [64]. Although preliminary, this collaborative specification underscores the importance of having input from clinicians and people with lived experience from the outset to maximise relevance, buy-in and value of new interventions [28].

The discussions uncovered a need not only for clinician training but also to address the significant structural barriers to APC care delivery such as a lack of access, availability and coordination of aphasia services, a lack of interdisciplinary working and organisational recognition of the importance of APC, and a lack of professional support and debriefing in keeping with earlier qualitative research with aphasia clinicians [26,35,38,65]. A need for effective data capture and monitoring was also evident; addressing such shortcomings is critical to complex intervention design [66] and to mitigate the impact of health inequities [67].

A key implementation priority was to focus on level 1, low-level interventions. Adapting and piloting existing psychosocial and self-management interventions, both aphasia specific and more general, were recommended. A wide range of recent intervention literature offers possibilities for further examination including Aphasia ASK (Action Success Knowledge) [68], Aphasia ADaPT (Aphasia, Depression, and Psychological Therapy) [12], "Kalmer" [69], and Aphasia PRISM (PRevention Intervention and Support in Mental health) [70]. Creative and mind-body interventions were of interest across both research groups. This is in keeping with a growing body of research on alternative and creative modality

psychological interventions [71], which may be advantageous in the context of aphasia, given their minimal language load [72]. The importance of individuals being able to choose the interventions they avail of was also proposed and resonates with contemporary literature. The Aphasia PRISM [70] is an individualised and protocolised intervention that offers people with aphasia a choice of therapies (behavioural activation, problem solving or relaxation).

One novel shift from the international stroke and aphasia stepped care literature was the suggestion that a 'level 0' might validate inherent personhood and potential for personally defined recovery and adjustment without the context of formal support and input. This was recognised to have synergies with the Personal Recovery mental health paradigm [61], which has previously been examined in the context of 'living well' with aphasia [54]. The inclusion of a conceptual Level 0 reflects stakeholder recognition that recovery often begins before and outside formal service engagement. Examples include peer support, informal befriending, social prescribing, and community activities that promote psychological well-being. These activities may be accessed directly, without referral, and may offer early benefits such as reduced isolation or increased confidence. While Level 0 does not involve clinical intervention, it may serve as a platform for identifying individuals who would benefit from more structured support. Engagement and outcomes at Level 0 could be monitored through attendance tracking, simple mood or wellbeing self-ratings, informal goal setting, or feedback from peers or family members. These low-burden and community-engaged approaches reflect the ethos of Level 0 and tend to be viewed by stakeholders as acceptable, valid and feasible in community-based and/or community-led practice [73]. Specific methods may include: (i) attendance records; (ii) pictorial mood scales or visual analogue tools co-developed with people with aphasia; (iii) Goal Attainment Scaling adapted for supported conversation; (iv) structured or informal feedback from peers, carers, or group leaders; and (v) brief reflective conversations around personally meaningful goals.

## Methodological critique

Every effort was taken to create authentic and inclusive research spaces for each group. All meetings incorporated PLA tools and techniques, which have been used successfully in aphasia PPI research [45,50], and in online co-design sessions with clinicians [49]. Meetings were initially convened online owing to the Covid pandemic; however, this resulted in 1 PPI contributor who was previously involved in research with the first author, opting out of this study. When in-person restrictions were lifted, the PPI contributors elected to meet in person again; this facilitated more engaged meetings, which supported more sustained, richer discussions compared with online meetings. Conversely, the online clinician meetings, necessitated by the pandemic, appeared to enhance efficiency, accessibility, and collaboration among clinicians. Such positives are reported elsewhere [74].

There are several important methodological limitations, and the findings should be appraised accordingly. This study provides some initial ideas generated through participatory research discussions with a small number of aphasia clinicians and PPI contributors. It is likely that these are not comprehensively representative of all services, settings, and experiences. The clinician group predominantly represents senior, specialised roles, whilst underrepresenting frontline primary care providers such as General Practitioners and community nurses. The aphasia group comprised only working-aged individuals living with aphasia for 10 years and longer, thereby excluding those with acute presentations and non-stroke aetiologies of aphasia, also populations with distinct psychological needs. This is particularly salient given that the agreed scope and focus of coordinated APC in this study ideally encompasses non-stroke aetiologies as well. It is important that the insights generated are appraised accordingly and that future research draws on sustained engagement with a wider range of interest holders, including families of people with aphasia, policy makers, and a wider range of key gatekeepers to mental health support in Ireland. Future research should also explore opportunities for combined meetings with the clinician co-researchers, aphasia PPI contributors and other interest holders. Furthermore, future participatory research in this area should give greater analytic weighting to knowledge generated with people with lived experience, in order to address the imbalance in the present study, which drew more heavily on clinician contributions.

In this study, NPT was used as an organising framework to explore APC in Ireland. Harnessing theory enhances research rigour and transferability so that the findings may inform future research in this area [75]. NPT constructs will underpin a future rapid realist review to generate a programme theory articulating context-mechanism-outcome configurations of how APC relating to non-progressive, acquired brain injury is expected to 'work' [76]. Component interventions, contextual implementation strategies, and outcome measures (clinical, service and economic) will be identified. This knowledge will be iteratively refined through participatory research with diverse interest holders representing lived experience, research, policy and practice, and used in the design and evaluation of new care pathways.

## Conclusion

Using a participatory implementation research methodology, people with aphasia and clinicians were engaged in research discussions on implementing a coordinated pathway of aphasia psychological care in Ireland, considering structural and local barriers, priority areas, and maximising cost and clinical efficiencies. This novel study demonstrates the potential of participatory approaches to inform APC design and highlights directions for future research.

While this study focused on Ireland, many of the structural barriers identified are shared across healthcare systems that experience regional fragmentation, limited psychological resources, or gaps in community integration. The principles developed through this participatory process, such as the importance of a shared vision, stepped care design, flexible referral routes, and integrated informal support, are likely to be applicable in other settings. In particular, jurisdictions with underdeveloped psychological support for people with aphasia or low interdisciplinary coordination may find value in the stakeholder-led methods and the emerging APC framework. The participatory implementation approach used here may also be adapted to other conditions or contexts, supporting locally relevant intervention planning without assuming direct replication.

## Supporting information

**S1 File. Standards for Reporting Qualitative Research (SRQR).doc.**
(DOCX)

**S2 File. Integrated summary and analysis of clinician meetings and contributions.pdf Integrated summary and analysis of clinician meetings and contributions.**
(PDF)

## Acknowledgments

We sincerely thank our aphasia PPI contributors for generously sharing their time and insights.

## Author contributions

**Conceptualization:** Molly Manning.

**Data curation:** Molly Manning.

**Formal analysis:** Molly Manning.

**Funding acquisition:** Molly Manning.

**Methodology:** Molly Manning, Aoife Carolan.

**Project administration:** Molly Manning.

**Validation:** Brian McClean, Libby Cunningham, Jolene Dervin, Marcia Ward, Aoife Carolan.

**Writing – original draft:** Molly Manning, Brian McClean, Libby Cunningham, Jolene Dervin, Marcia Ward, Aoife Carolan.

**Writing – review & editing:** Molly Manning, Brian McClean, Libby Cunningham, Jolene Dervin, Marcia Ward, Aoife Carolan.

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
