## [Editor Report · Decision Letter 0]

20 Nov 2025

Dear Dr. Manning,

Thank you for submitting your manuscript to PLOS ONE. After careful consideration, we feel that it has merit but does not fully meet PLOS ONE’s publication criteria as it currently stands. Therefore, we invite you to submit a revised version of the manuscript that addresses the points raised during the review process.

We look forward to receiving your revised manuscript.

Kind regards,

Rong Yan

Academic Editor

PLOS ONE

Journal Requirements:

This work was supported by Seed Funding from the Faculty of Education and Health Sciences, University of Limerick, awarded to Dr Molly Manning in 2021.

3. We noted in your submission details that a portion of your manuscript may have been presented or published elsewhere. Please clarify whether this publication was peer-reviewed and formally published. If this work was previously peer-reviewed and published, in the cover letter please provide the reason that this work does not constitute dual publication and should be included in the current manuscript.

Additional Editor Comments:

1. Significance of the Research

This manuscript addresses a critical and understudied gap in clinical practice: the absence of coordinated, context-specific psychological care for individuals with aphasia within the Irish healthcare system. The well-documented association between aphasia and adverse psychosocial outcomes, including depression, anxiety, and social isolation, underscores the urgency of this issue. The study's commitment to stakeholder co-design—centering the perspectives of people with aphasia (PWA) and interdisciplinary clinicians—aligns directly with PLOS ONE's mission to publish applied health research with tangible real-world impact.

The relevance of this work is further enhanced by its grounding in the distinctive Irish healthcare context, with explicit reference to pertinent national policies such as the National Stroke Strategy 2022–27 and the Sláintecare reform program. By systematically linking local service deficiencies (e.g., constrained access to psychological services, inconsistent mood screening protocols) to broader implementation science challenges, the study occupies a unique niche in the extant literature, which frequently focuses on larger healthcare systems while neglecting smaller, contextually specific environments like Ireland.

Notwithstanding these strengths, the manuscript would benefit from a more explicit articulation of the transferability of its findings beyond the local context. For instance, the identified principles of a "unified vision" and "flexible referral pathways" may yield valuable lessons for other jurisdictions grappling with comparable resource limitations or decentralized service delivery models. Elaborating on this dimension would significantly bolster the manuscript's appeal to PLOS ONE's international readership and reinforce its broader scholarly significance.

The manuscript's principal innovative contribution lies in its methodological approach: the deliberate integration of Participatory Learning and Action (PLA) tools with Normalization Process Theory (NPT) to structure stakeholder engagement. While both PLA and NPT have been employed independently within aphasia research, their synergistic combination here is both deliberate and purposeful. PLA facilitates inclusive knowledge co-creation with PWA, a population often facing significant barriers to research participation, while NPT supplies a robust theoretical framework to translate these insights into implementable strategies. This dual methodology effectively addresses a recognized limitation in prior participatory studies, which often lack explicit connections to implementation science theory.

The proposition of a "level 0" within the stepped care model—formally acknowledging the informal, non-clinical support systems upon which PWA frequently rely (e.g., peer networks, self-management strategies)—constitutes another substantive contribution. Although the Personal Recovery paradigm has previously informed aphasia care, the manuscript's focus on operationalizing "level 0" as an integral component of stepped care, rather than a peripheral consideration, represents a novel advancement. This approach validates the agency of PWA and recognizes that recovery processes often originate outside formal clinical settings—a conceptual shift with potential to reconfigure stepped care design for neurogenic disorders beyond aphasia.

To fully realize this innovative potential, the manuscript should elaborate on the functional interplay between "level 0" and established care levels. Specifically, it should address operational considerations such as: What mechanisms would trigger referrals from level 0 to level 1? How would engagement or outcomes at level 0 be assessed in clinical practice? Providing concrete examples or stakeholder-derived operational guidelines would substantially enhance the credibility of "level 0" as an actionable clinical concept.

2. Research Methods

The methodological framework is generally robust and adheres to PLOS ONE's standards for qualitative inquiry, evidenced by compliance with the Standards for Reporting Qualitative Research (SRQR) and clear documentation of ethical approval (University of Limerick, 2021_12_19). The decision to conduct separate workshops for clinicians and PWA, followed by structured cross-group knowledge integration, is particularly commendable. This approach enables candid discourse within each stakeholder group while ensuring the systematic incorporation of diverse perspectives—a crucial safeguard against tokenistic patient and public involvement.

However, three significant limitations require remediation to satisfy PLOS ONE's rigor requirements:

Sample Size and Diversity

The participant cohort (6 clinicians, 3 PWA), while permitting in-depth participatory dialogue, exhibits limited diversity across critical dimensions. The clinician sample overrepresents senior, specialized roles (e.g., Senior Speech and Language Therapists, Clinical Neuropsychologists) while underrepresenting frontline primary care providers (e.g., General Practitioners, community nurses) who frequently function as gatekeepers to aphasia psychological care. Regarding PWA participants, all were working-aged individuals with long-term aphasia (10–15 years post-stroke), thereby excluding those with acute presentation or non-stroke etiologies (e.g., traumatic brain injury)—populations with distinct psychological needs. The authors should explicitly acknowledge these limitations and, where feasible, supplement the discussion with insights from relevant literature concerning underrepresented groups to better contextualize their findings.

Transparency in Data Analysis

Although the manuscript describes the employment of PLA techniques (e.g., sticky note brainstorming, card-sorting) and NPT operationalization, it lacks sufficient detail regarding systematic analytical procedures. Critical methodological information is absent, including: The specific coding methodology applied to qualitative data (e.g., inductive versus deductive approaches); Measures implemented to ensure coding reliability (e.g., independent coding by multiple researchers, consensus-building procedures); The precise application of NPT constructs to interpret emergent themes. The inclusion of a supplementary table or detailed paragraph elucidating the analytical workflow would satisfy PLOS ONE's requirement for reproducible qualitative methodologies.

Data Availability

The current data availability statement ("No - some restrictions will apply") fails to comply with PLOS ONE's policy mandate that authors "make all data underlying the findings described fully available, without restriction" (except where legally or ethically justified). The authors must provide explicit clarification regarding:

The specific categories of data subject to restrictions (e.g., raw verbatim comments to protect participant anonymity);

Accessibility procedures for non-restricted data (e.g., de-identified thematic summaries, workshop protocols);

The ethical or legal rationale underpinning any restrictions (e.g., specific informed consent agreements with PWA).

A comprehensive and policy-compliant data availability statement is mandatory for resubmission.

3. Writing and Presentation

The manuscript demonstrates sound organizational structure and accessibility, with a coherent progression from background through methods, results, and discussion. Notable strengths include: clear explication of technical terminology (e.g., "stepped care," "PLA") appropriate for interdisciplinary readership; strategic incorporation of direct quotations from PWA, anchoring findings in lived experience; consistent alignment between stated research objectives and presented results. Nevertheless, several revisions would enhance clarity and professional presentation:

Redundancy

Several key points (e.g., the necessity for enhanced clinician training, barriers to interdisciplinary collaboration) are repetitively addressed across the introduction, results, and discussion sections. Maybe the authors could further streamline the similar expressions to improve readability.

Table Design

Tables 1 (Operationalisation of NPT Constructs) and 2 (Sequence and Focus of Meetings) require enhanced detail to ensure methodological reproducibility:

Table 1 should incorporate concrete illustrations of how NPT constructs guided discussions (e.g., "Context: Prompted clinicians to examine how HSE funding constraints impact APC delivery");

Table 2 should rectify formatting inconsistencies (e.g., truncated prompts) and incorporate temporal markers (e.g., "April 2022: Clinicians 1 workshop") to clarify the research timeline.

Reference Consistency

The reference list contains several technical inaccuracies:

Baker (20) (cited in Section 1.46) is omitted from the references;

Citations for Leamy et al. (2011) and Manning et al. (2019) appear in incorrect sequence.

Rectifying these errors is necessary to ensure compliance with PLOS ONE's bibliographic standards.

4. Conclusion

This manuscript constitutes a valuable contribution to aphasia care research through its principled centering of stakeholder perspectives and its systematic linkage of participatory insights with actionable implementation strategies. Its focused examination of the Irish healthcare context addresses a significant literature gap, while its methodological integration of PLA and NPT offers a replicable framework for future co-design studies. Provided the authors address the identified concerns regarding sample diversity, analytical transparency, and presentational clarity, the revised manuscript will fully satisfy PLOS ONE's academic standards and serve as an important resource for clinicians, policymakers, and researchers dedicated to enhancing psychological support for people with aphasia. To facilitate efficient re-evaluation, the authors should prioritize the following revisions:

1) Elaborate the discussion regarding the adaptability of findings to other healthcare contexts to enhance international relevance.

2) Provide concrete operational specifications for the "level 0" stepped care model, including potential referral triggers and assessment approaches.

3) Incorporate a comprehensive description of the data analysis workflow, detailing coding methodologies and reliability measures.

4) Revise the data availability statement to achieve full compliance with PLOS ONE policies, explicitly clarifying restrictions and access procedures.

---

## [Author Response · Author response to Decision Letter 1]

18 Dec 2025

A 'response to reviewers' doc has been uploaded.

---

## [Decision Letter · Decision Letter 1]

10 Mar 2026

Dear Dr. Manning,

Thank you for submitting your manuscript to PLOS ONE. After careful consideration, we feel that it has merit but does not fully meet PLOS ONE’s publication criteria as it currently stands. Therefore, we invite you to submit a revised version of the manuscript that addresses the points raised during the review process.

Kind regards,

Rong Yan

Academic Editor

PLOS One

**Journal Requirements:**

**Additional Editor Comments:**

In light of the comments and feedback provided by the two reviewers, I find that your revisions have been carried out satisfactorily, and I am pleased to recommend this manuscript for publication. Prior to submitting the final version, you still need to further revise the paper in accordance with the remaining suggestions on method and discussion sections from the two reviewers. Congratulations on your work, and thank you for your valuable contribution.

Reviewers' comments:

Reviewer's Responses to Questions

**Comments to the Author**

Reviewer #2: All comments have been addressed

Reviewer #3: All comments have been addressed

2. Is the manuscript technically sound, and do the data support the conclusions?

Reviewer #2: Yes

Reviewer #3: Yes

3. Has the statistical analysis been performed appropriately and rigorously?

Reviewer #2: Yes

Reviewer #3: Yes

4. Have the authors made all data underlying the findings in their manuscript fully available?

PLOS Data policy  requires authors to make all data underlying the findings described in their manuscript fully available without restriction, with rare exception (please refer to the Data Availability Statement in the manuscript PDF file). The data should be provided as part of the manuscript or its supporting information, or deposited to a public repository. For example, in addition to summary statistics, the data points behind means, medians and variance measures should be available. If there are restrictions on publicly sharing data—e.g. participant privacy or use of data from a third party—those must be specified.

Reviewer #2: Yes

Reviewer #3: Yes

5. Is the manuscript presented in an intelligible fashion and written in standard English?

Reviewer #2: Yes

Reviewer #3: Yes

6. Review Comments to the Author

**Reviewer #2:**  Thank you for your comprehensive revisions, which have addressed the previous concerns effectively. Your manuscript is now much stronger. I have only a few minor, optional suggestions for fine-tuning the text before final publication. Thank you for your comprehensive revisions, which have addressed the previous concerns effectively. Your manuscript is now much stronger. I have only a few minor, optional suggestions for fine-tuning the text before final publication.

In the “Knowledge generation and synthesis” subsection, it might be helpful to add a brief sentence explaining how the iterative thematic analysis complemented the NPT-guided discussion prompts. This would further clarify your analytical strategy for readers.

Regarding the discussion on creative therapies, the evidence is currently described as “equivocal.” To maintain a strong yet perfectly balanced recommendation, you might consider phrasing that acknowledges the developing evidence base while underscoring the stakeholder-endorsed rationale for its inclusion.

These are minor points aimed at final polishing. Congratulations on a well-executed and insightful study. I believe it will be of interest to the journal’s readership.

**Reviewer #3:**  This manuscript addresses an important and underdeveloped area: the participatory development and implementation planning of psychological care for people with aphasia in Ireland. The integration of Participatory Learning and Action (PLA) methods with Normalisation Process Theory (NPT) provides a coherent and contextually grounded framework. The authors have responded carefully to prior reviewer comments, and the manuscript has improved substantially in clarity, structure, and transparency. I just have some minor points concerning the Section of Materials and methods. This manuscript addresses an important and underdeveloped area: the participatory development and implementation planning of psychological care for people with aphasia in Ireland. The integration of Participatory Learning and Action (PLA) methods with Normalisation Process Theory (NPT) provides a coherent and contextually grounded framework. The authors have responded carefully to prior reviewer comments, and the manuscript has improved substantially in clarity, structure, and transparency. I just have some minor points concerning the Section of Materials and methods.

1) In the Methods section, the authors have expanded the description of the analytic process. This revision is helpful. However, it is still not fully clear whether the thematic analysis was primarily inductive, deductive (guided by NPT), or hybrid. Please specify this explicitly.

2) In the same section, the manuscript states that the first author led the coding process with ongoing scrutiny from co-authors. It would strengthen methodological transparency to clarify whether any independent coding of raw data was conducted by more than one researcher before consensus discussions took place.

3) Also in the Methods section, the manuscript notes that PPI meeting data were not formally analysed thematically but were grouped at a meta-level. Please provide a clearer methodological justification for this difference in analytic treatment and explain how this approach ensured that PPI contributions were given equal analytic weight.

7. PLOS authors have the option to publish the peer review history of their article (what does this mean? ). If published, this will include your full peer review and any attached files.). If published, this will include your full peer review and any attached files.

**Do you want your identity to be public for this peer review?**  For information about this choice, including consent withdrawal, please see our  For information about this choice, including consent withdrawal, please see our Privacy Policy .

Reviewer #2: **Yes:** Huichao Bi, School of Education, Tsinghua UniversityHuichao Bi, School of Education, Tsinghua University

Reviewer #3: **Yes:** Baoshan ZhangBaoshan Zhang

---

## [Author Response · Author response to Decision Letter 2]

12 Mar 2026

RESPONSE TO REVIEWERS

Reviewer 2 Comments

Changes made

1

In the “Knowledge generation and synthesis” subsection, it might be helpful to add a brief sentence explaining how the iterative thematic analysis complemented the NPT-guided discussion prompts. This would further clarify your analytical strategy for readers.

Thank you for this suggestion.

We have overhauled the relevant content in the Knowledge Generation and Synthesis section as follows:

“Initially, NPT informed the development of questions related to clinicians’ sense‑making, their perceptions and experiences on current care (Differentiation, Individual Specification). After each clinician meeting, all contributions and virtual ‘stickies’ were systematically reviewed to identify similarities, differences, and emerging insights. These informed the development of preliminary themes and descriptors capturing patterns of meaning relevant to the research questions.

A summary of these evolving themes, supported by all contributions, was presented at the start of each subsequent meeting, functioning as a de facto data analysis clinic This approach enabled member checking and strengthened trustworthiness (59). The emerging themes guided further discussion about principles of good care should entail, and the perceived value and differences compared with current practice (Communal Specification and Appraisal).

The knowledge generated through these discussions was again synthesised and thematically organised. Refined themes were subsequently presented to guide dialogue on stepped‑care design and priorities for implementation (Cognitive Participation and Collective Action constructs). Following the second clinician meeting, these evolving themes were also presented to PPI contributors, whose input was incorporated alongside clinician comments to validate findings and inform subsequent meetings.

2 Regarding the discussion on creative therapies, the evidence is currently described as “equivocal.” To maintain a strong yet perfectly balanced recommendation, you might consider phrasing that acknowledges the developing evidence base while underscoring the stakeholder-endorsed rationale for its inclusion. Thank you for this helpful suggestion – we have simplified the wording as follows:

“Creative and mind-body interventions were of interest across both research groups. This is in keeping with a growing body of research on alternative and creative modality psychological interventions (71), which may be advantageous in the context of aphasia, given their minimal language load

(72).”

Reviewer 3 Comments

Changes made

1 In the Methods section, the authors have expanded the description of the analytic process. This revision is helpful. However, it is still not fully clear whether the thematic analysis was primarily inductive, deductive (guided by NPT), or hybrid. Please specify this explicitly. Thank you for these helpful comments.

The Knowledge Generation and Synthesis section has been revised for clarity relating to (1) inductive analysis process; (2) independent coding by first author; (3) analysis of PPI contributions. A new sentence has also been added to the Methodological Critique relating to the imbalance in analytical weight across PPI and clinician contributions.

REVISED TEXT IN KNOWLEDGE GENERATION & SYNTHESIS:

“Although NPT constructs guided the discussion prompts, the knowledge generated through PLA techniques was analysed using an iterative thematic analysis process (58). The first author led this process, coding all material, with continuous scrutiny and collaborative input from co-authors during meetings and through co-creation of the final report. The analysis was inductive and highly iterative, with themes refined through ongoing discussion.

Initially, NPT informed the development of questions related to clinicians’ sense‑making, their perceptions and experiences on current care (Differentiation, Individual Specification). After each clinician meeting, all contributions and virtual ‘stickies’ were systematically reviewed to identify similarities, differences, and emerging insights. These informed the development of preliminary themes and descriptors capturing patterns of meaning relevant to the research questions.

A summary of these evolving themes, supported by all contributions, was presented at the start of each subsequent meeting, functioning as a de facto data analysis clinic This approach enabled member checking and strengthened trustworthiness (59). The emerging themes guided further discussion about principles of good care should entail, and the perceived value and differences compared with current practice (Communal Specification and Appraisal).

The knowledge generated through these discussions was again synthesised and thematically organised. Refined themes were subsequently presented to guide dialogue on stepped‑care design and priorities for implementation (Cognitive Participation and Collective Action constructs). Following the second clinician meeting, these evolving themes were also presented to PPI contributors, whose input was incorporated alongside clinician comments to validate findings and inform subsequent meetings.

To enhance reflexivity, the first author maintained detailed notes, documenting emerging interpretations, positionality, and potential biases (60). A de-identified version of the final clinician slide deck—incorporating all contributions and the outputs of the iterative thematic analysis—is provided in Supplementary File S2. Feedback from this final stage was thematically summarised, forming the basis of the final reported themes.”

REVISED TEXT IN METHODOLOGICAL CRITIQUE:

“Furthermore, future participatory research in this area should give greater analytic weighting to knowledge generated with people with lived experience, in order to address the imbalance in the present study, which drew more heavily on clinician contributions.”

2 In the same section, the manuscript states that the first author led the coding process with ongoing scrutiny from co-authors. It would strengthen methodological transparency to clarify whether any independent coding of raw data was conducted by more than one researcher before consensus discussions took place.

3 Also in the Methods section, the manuscript notes that PPI meeting data were not formally analysed thematically but were grouped at a meta-level. Please provide a clearer methodological justification for this difference in analytic treatment and explain how this approach ensured that PPI contributions were given equal analytic weight.

---

## [Decision Letter · Decision Letter 2]

25 Mar 2026

Advancing aphasia psychological care in Ireland: A Participatory Study with people with aphasia and clinicians

PONE-D-25-43919R2

Dear Dr. Manning,

Thank you for your careful attention to all the suggestions made during the review process. We’re pleased to inform you that your manuscript has been judged scientifically suitable for publication and will be formally accepted for publication once it meets all outstanding technical requirements.

Kind regards,

Antony Bayer

Academic Editor

PLOS One

Additional Editor Comments (optional):

Reviewers' comments:

Reviewer's Responses to Questions

**Comments to the Author**

Reviewer #3: All comments have been addressed

2. Is the manuscript technically sound, and do the data support the conclusions?

Reviewer #3: Yes

3. Has the statistical analysis been performed appropriately and rigorously?

Reviewer #3: Yes

4. Have the authors made all data underlying the findings in their manuscript fully available?

Reviewer #3: Yes

5. Is the manuscript presented in an intelligible fashion and written in standard English?

Reviewer #3: Yes

Reviewer #3: The authors have responded carefully to my comments and made appropriate revisions to the manuscript. I appreciate the efforts made to improve the paper. I have no additional comments and recommend acceptance of the manuscript.

**Do you want your identity to be public for this peer review?** For information about this choice, including consent withdrawal, please see our For information about this choice, including consent withdrawal, please see our Privacy Policy .

Reviewer #3: No

---

## [Editor Report · Acceptance letter]

PONE-D-25-43919R2

PLOS One

Dear Dr. Manning,

I'm pleased to inform you that your manuscript has been deemed suitable for publication in PLOS One. Congratulations! Your manuscript is now being handed over to our production team.

Kind regards,

on behalf of

Professor Antony Bayer

Academic Editor

PLOS One